# miRNAs’ Cross-Involvement in Skin Allergies: A New Horizon for the Pathogenesis, Diagnosis and Therapy of Atopic Dermatitis, Allergic Contact Dermatitis and Chronic Spontaneous Urticaria

**DOI:** 10.3390/biomedicines11051266

**Published:** 2023-04-24

**Authors:** Raffaele Brancaccio, Giuseppe Murdaca, Rossella Casella, Teresa Loverre, Laura Bonzano, Eustachio Nettis, Sebastiano Gangemi

**Affiliations:** 1Dermatology Unit, Azienda Unità Sanitaria Locale-IRCCS di Reggio Emilia, 42122 Reggio nell’Emilia, Italy; 2Department of Internal Medicine, University of Genova, 16132 Genova, Italy; 3Department of Internal Medicine, Ospedale Policlinico San Martino, 16132 Genova, Italy; 4Department of Emergency and Organ Transplantation, School of Allergology and Clinical Immunology, University of Bari Aldo Moro, Policlinico di Bari, 70124 Bari, Italy; 5Department of Clinical and Experimental Medicine, School and Division of Allergy and Clinical Immunology, University of Messina, 98125 Messina, Italy

**Keywords:** miRNA, atopic dermatitis, chronic spontaneous urticaria, allergic contact dermatitis, antagomirs, cytokines, inflammation, skin, pathogenesis, treatment

## Abstract

Skin inflammation is a common underlying feature of atopic dermatitis, allergic contact dermatitis and chronic spontaneous urticaria. The pathogenetic mechanisms have not been fully elucidated. The purpose of this study was to examine whether miRNA, by regulating inflammatory mechanisms through the modulation of innate and adaptive immune responses, could play a major role in the pathogenesis of these skin conditions. We conducted a narrative review using the Pubmed and Embase scientific databases and search engines to find the most relevant miRNAs related to the pathophysiology, severity and prognosis of skin conditions. The studies show that miRNAs are involved in the pathogenesis and regulation of atopic dermatitis and can reveal an atopic predisposition or indicate disease severity. In chronic spontaneous urticaria, different miRNAs which are over-expressed during urticaria exacerbations not only play a role in the possible response to therapy or remission, but also serve as a marker of chronic autoimmune urticaria and indicate associations with other autoimmune diseases. In allergic contact dermatitis, miRNAs are upregulated in inflammatory lesions and expressed during the sensitization phase of allergic response. Several miRNAs have been identified as potential biomarkers of these chronic skin conditions, but they are also possible therapeutic targets.

## 1. Introduction

Atopic dermatitis, allergic contact dermatitis and chronic spontaneous urticaria are common inflammatory conditions involving the skin. This inflammation may result from unique and, yet, shared pathogenic mechanisms, which may account for their coexistence and association in atopic patients.

### 1.1. Pathogenesis of Atopic Dermatitis

Atopic dermatitis (AD) is the most common inflammatory skin disorder in the developed world, with a lifetime prevalence of 15–20% [1]. It is often associated with other atopic conditions, such as asthma, allergic rhinitis and chronic rhinosinusitis with nasal polyps [2]. AD is a chronic, relapsing and inflammatory skin disease characterized by intense itching and excoriations with erythematous, xerotic, fissured and lichenificated skin, as well as an increased risk of skin infections [3,4]. The pathogenesis of AD has not been completely elucidated. Genetic heritage and its interactions with the environment contribute to epidermal skin barrier disruption, altered inflammatory response and skin microbiome dysbiosis [5,6], and oxidative stress also plays an important pathogenetic role [7]. Recent studies have pointed to the key role of epigenetic changes in AD development, mainly mediated by DNA methylation, histone acetylation and the action of specific non-coding RNAs, including micro-RNAs (miRNAs), small interfering RNAs, long non-coding RNAs and Pivi-interacting RNAs [8]. Skin barrier abnormalities include decreased filaggrin, loricrin, involucrin, ceramides, antimicrobial peptides and disorders of tight junctions [9,10]. In particular, filaggrin, a structural protein responsible for keratinization, moisturization and antimicrobial peptide functions, is mutated in 15–50% of patients with AD [11,12]. Skin barrier disruption stimulates keratinocytes and dendritic cells to produce activation-regulated chemokine (TARC), thymic stromal lymphopoietin (TSLP) and cytokines, including IL-1β, IL-25 and IL-33, which trigger type-2 inflammation, activating both innate and adaptative immunity [13,14]. Type-2 innate lymphoid cells and Th2 cells produce IL-4, IL-5 and IL-13 [13], key mediators of type-2 response. They promote Th-naïve lymphocytes developing Th2 phenotypes, B lymphocyte isotypes switching from IgM to IgE [15], eosinophil recruitment and activation [16], itch exacerbation [17] and skin remodeling toward fibrosis in the chronic disease phase [1].

### 1.2. Pathogenesis of Allergic Contact Dermatitis

Allergic contact dermatitis (ACD) is a common inflammatory skin disorder characterized by pruritus, erythema, vesicles and scaling of the skin [18]. ACD is common, with some studies demonstrating prevalence rates as high as 20% in the general population, and it accounts for the vast majority of occupational skin disorders in the Western world [18,19]. ACD involves type IV-mediated hypersensitivity to a specific allergen, resulting in an inflammatory response with exposure. The first phase is sensitization, when a person is first exposed to a hapten, which is defined as a low-molecular-weight antigen that, when bound to a larger carrier, can elicit an immune response.

After hapten has been recognized and phagocytized by dermal dendritic cells and Langerhans cells, the hapten–peptide complexes migrate to regional lymph nodes and induce the proliferation and circulation of hapten-specific T cells (Th1, Th2, Th17 and T regulatory cells) and the creation of effector and memory T cells. Clinical symptoms of ACD are produced in the elicitation phase, when re-exposure to the allergen, recognized by the sensitized hapten-specific T cells, causes an inflammatory cascade of cytokines and cellular infiltrates [20]. Since the reaction to the allergen is not always immediate, with a possible delay of up to 72 h, it may be difficult to identify the culprit agents. Thorough questioning of the patient’s occupation, hobbies and any changes in personal products or clothing is helpful [21], and certain distributions, such as on the eyelid, lateral face, central face, neck or hands suggest the consideration of ACD. Patch testing is the only practical, scientific and objective method to confirm the diagnosis of ACD.

### 1.3. Pathogenesis of Chronic Spontaneous Urticaria

Chronic spontaneous urticaria (CSU) is defined as the spontaneous appearance of itchy red wheals and/or angioedema for more than six weeks without any apparent cause [22,23], and it affects about 1.8% of the adult population [24]. CSU is mainly caused by the activation of cutaneous mast cells (MCs), leading to the release of histamine and other mediators, such as platelet-activating factor (PAF) and cytokines, which results in vasodilatation, plasma extravasation and sensory nerve activation, as well as cell recruitment in the urticarial lesions (mainly T cells, eosinophils and basophils). Wheals are characterized by edema of the upper and mid-dermis, whereas similar changes may occur in the lower dermis, resulting in angioedema [22]. The pathophysiology of CSU is not well understood, but there are two main pathogenetic mechanisms underlying the disease: dysregulation of intracellular signaling pathways within mast cells and basophils, which leads to defects in the trafficking or function of these cells, and development of autoantibodies to FcεRIα or IgE on both mast cells and basophils [25]. Basophil numbers appear to be reduced in at least 50% of patients [26], and they seem to be hyporeactive [27]. Furthermore, their development is dependent on the interaction of stem cell factor (SCF) with the c-kit, as well as TGF-β produced by Tregs. In CSU, circulating Tregs (CD4 + CD25 + FOXP3+) are reduced and/or defective compared with normal cells [28], and the reduced frequency of Tregs is consistent with an autoimmune hypothesis for CSU [29,30]. These events lead to mast cell degranulation and the predisposition to pathological mast cell activation if inappropriately upregulated. However, autoimmune theory is the most widely accepted hypothesis for explaining the pathogenesis of CSU, as it is thought that up to 45% of cases of CSU have autoimmune etiology [25]. Specifically, CSU is associated with the development of circulating IgG antibodies against IgE and the high-affinity IgE receptor FcεR1; approximately 40% of these patients have circulating antibodies against one of the two targets [31], with a higher frequency of positivity in CSU patients with a positive autologous serum skin test (ASST) [32]. It seems likely that different pathomechanisms are interlinked in CSU rather than independent cascades [33].

### 1.4. MicroRNAs

miRNAs represent an abundant class of small non-coding RNAs which regulate gene expression post-transcriptionally [34]. miRNAs are short (20–22 nucleotides long), endogenous and single-stranded RNAs, whose genes [35], highly conserved throughout evolution across species [36], are located within the introns and exons of protein-coding genes [37].

The first miRNAs were found in Caenorhabditis elegans in early 1990s [34]; afterward, the first human miRNA was discovered in the 2000s [38]. Currently, 2588 mature miRNAs have been identified for humans in miRbase [39], and it is estimated that they modulate more than 60% of all human protein-coding genes by sequence-specific base pairing [35]. In fact, it is widely recognized that one miRNA may regulate many genes as its targets, while one gene may be targeted by many miRNAs. [40] 

Primary transcripts of miRNAs are generated by RNA polymerase II and sequentially processed by ribonuclease III class enzymes [37] (Drosha) to produce an miRNA precursor of 70 nucleotides (the pre-miRNA) [41]. After nuclear processing, the pre-miRNA is transferred to the cytosol, where Dicer, another ribonuclease III class enzyme, cleaves it into mature double-stranded miRNA with 19–24 nucleotides. One strand of this mature miRNA duplex is then incorporated into the RNA-induced silencing complex (RISC) [42] and serves as a guide for the recognition of target mRNAs [43]. The other strand is unrolled from the guide strand and degraded, but, in some cases, both strands of the miRNA duplex are functional [44]. The miRNA–RNA-induced silencing complexes interact with mRNA targets by sequence-specific base pairing, generally within the 3′-untranslated region of the target mRNAs [45]. Two different mechanisms of RISC-mediated gene regulation exist, and they depend on the level of miRNA–mRNA complementarity: at sites with extensive complementarity, the miRNA can mediate mRNA cleavage; however, sites with a lower degree of complementarity may lead to translational repression or mRNA destabilization [46].

Circular RNAs (CircRNAs) are single-stranded, covalently closed RNA molecules that are ubiquitous across species ranging from viruses to mammals [47]. One of the most commonly reported mechanisms of action of circRNAs is the “sponging” of miRNAs by binding to the miRNA response element (MRE), thereby indirectly increasing the transcription of their target mRNAs [48].

MiRNAs play an important role in biological processes such as cellular proliferation, differentiation and apoptosis. Further, they are implicated in inflammatory diseases, cancer [49], immune response, neural development, DNA repair and oxidative stress response [50].

In addition, miRNAs are stable in body fluids, so they are more available for assays and less invasive than biopsies [51]. Thus, the interest in miRNAs as both biomarkers and possible therapeutic targets is increasing [52].

The aim of this study is to collect and review data from the current literature on the involvement of miRNA in AD, ACD and CSU.

The bibliographic search for our review was conducted using Pubmed and Embase. The keywords selected for our searching process were “miRNAs” and “microRNAs” combined with “atopic dermatitis”, “urticaria” and “allergic contact dermatitis”. In our review, we included all the research articles indexed in peer-reviewed scientific journals that reported the role of miRNA in these three allergic skin disorders. From the eligible articles, data were extracted and cross-matched to find the most relevant and commonly expressed miRNAs reported in the literature that were correlated to the pathophysiology, severity and prognosis of AD, ACD and CSU.

## 2. miRNA in AD, ACD and CSU: Pathogenetic Role and Therapeutic Strategies

Increasing evidence suggests that miRNAs may play important roles in regulating physiological skin processions, such as self-renewal and differentiation, in stem cells. The most highly expressed miRNAs in normal-condition skin are miR-152, miR-143, miR-126, miR -21, miR-27a, miR-214, miR-16, miR-203, miR 125b, miR-34a, miR-205, miR-27b, miR-30b, miR-125a, miR-191, the miR-200 family (-200a, -200b, -200c, -141, -429), the miR-199 family (-199a, -199b) and the miR-19/-20 family (-19b, -20, -17-5p, -93) [53]. Skin inflammation is a common underlying feature of AD, ACD and CSU. However, the pathogenetic mechanisms at the base of these three skin conditions have not yet been fully elucidated. The results of our review suggest that miRNA, by regulating inflammatory mechanisms through the modulation of innate and adaptive immune responses, could play a major role in the pathogenesis of these three skin conditions.

### 2.1. Pro-Inflammatory and Anti-Inflammatory miRNA in AD

Several miRNAs have been found to be implicated in the crosstalk between inflammatory cells and keratinocytes in patients affected by atopic dermatitis (Figure 1). In particular, Sonkoly et al. [54] reported that mir-155 is one of the most significantly upregulated miRNAs in the lesional skin of patients affected by AD, suggesting its major role in the pathogenesis of the disease. MiR-155 is a multifunctional miRNA that is involved in immune cell maturation and the regulation of innate and adaptive immune systems, playing a critical role in both T- and B-cell responses [55,56,57]. Overexpression of miR-155 decreases CTLA-4 levels and increases the proliferation of T-helper cells, promoting chronic skin inflammation. Rebane et al. [58] demonstrated upregulation of miR-146 in the serum and skin of patients with AD compared to a healthy population, as well as in murine skin specimens obtained from an AD-like mice model in comparison to a control group.

MiR146a is a known anti-inflammatory and NFκB pathway-dependent miRNA expressed in B cells, T cells, monocytes and dendritic cells [59,60,61]. Its role is to regulate innate and adaptive immunity, regulating antibody production, inflammatory factor secretion and immune cell differentiation [62].

MiR-151a was also found to be overexpressed in the plasma of 500 atopic dermatitis patients by Chen XF et al. [63], whereas its expression in eczematous skin was not determined to be relevant. Upregulation of miR-151a favors the shift toward Th2 cell response, being predominant in AD [64] and suppressing the expression of Th1 cytokines including IL-2, IL-12 and INF-gamma [65].

According to Jia HZ et al. [66], differential miR-223 expression in the plasma of AD patients is correlated with the severity of the disease. More specifically, plasma miR223 expression levels in patients with severe AD were found to be significantly higher than in those with mild disease, urticarial patients and healthy volunteers. This correlation means that mir-223 could be a biomarker of the severity of the disease. MiR-223 is prevalently expressed by neutrophils, monocytes and eosinophils associated with cigarette smoking [67]. Herberth et al. [68] demonstrated that maternal and cord blood miR-223 expression was inversely proportional to Treg cell numbers. Lower Treg cell numbers at birth have been shown to increase the risk of AD in children.

According to Lv Y et al. [69], miR-483-5p was found to be upregulated in the serum of children with AD compared with controls. Further, the level of miR-483-5p in serum was significantly associated with other atopic conditions, such as rhinitis and asthma [70]. miR-483-5p has been identified as a fibrogenesis modulator, playing a role in collagen homeostasis.

Vaher H et al. [71] observed that miR-10a-5p was upregulated in both the lesional and non-lesional skin of 10 children with AD, leading to the hypothesis that increased expression of miR-10a-5p in the skin of AD patients might suppress keratinocyte proliferation. In fact, the upregulation of miR-10a-5p in skin was induced by atopic dermatitis-related cytokines, such as IL-4, IL-17 and IL-1β, which are able to reduce the expression of the proliferation marker Ki-67. In addition, miR-10a-5p impairs the normal epidermal barrier function by directly targeting hyaluronan synthase 3 (HS3), a damage-associated positive regulator of keratinocyte proliferation and migration.

Gu Chaoying et al. [72] observed that miR-29b was one of the most significantly upregulated miRNAs in the skin lesions of patients with AD as compared with healthy controls. In addition, the upregulation of miR-29b has been statistically associated with the development of AD. miR-29b-mediated expression of BCL2L2 is involved in the IFN-regulated cell proliferation and apoptosis of keratinocytes [72].

Lv Y et al. [69] also demonstrated the differential expression of miR-203 in diverse biologic specimens obtained from patients diagnosed with AD. Interestingly, miR-203 is upregulated both in serum and skin, whereas it is downregulated in urine samples [69,73]. Moreover, in children with AD, elevated miR-203 expression has been found to be significantly associated with TNFRI and TNFRII, suggesting a potential role of miR-203 in regulating these two inflammatory factors [69]. The main target gene of miR-203 has been found to be the regulator of cytokine production SOCS-313 (suppressor of cytokine signaling 3) [74].

MiRNA-143 has been shown to decrease IL-13 activity and inflammation by targeting and downregulating IL-13Rα1 in epidermal keratinocytes, thus playing a potential role in decreasing AD-induced skin inflammation. In detail, miR-143 opposes the negative regulation of filaggrin, loricrin and involucrin induced by IL-13 in human keratinocytes, potentially improving epidermal barrier function [75].

MiR-124 has been demonstrated to directly target nuclear factor (NF)-κB in B-cell lymphoma [76]. Yang et al. [77] investigated the role of miR-124 in atopic dermatitis, demonstrating that miR-124 expression was downregulated in chronic AD skin lesions. Tumor necrosis factor (TNF)-α and IFN-γ were also able to inhibit MiR-124 expression. Modulating the NF-κB pathway, miR-124 reduces chronic skin inflammation and inflammatory responses in keratinocytes in AD [77]. miRNA in atopic dermatitis are described in Table 1.

### 2.2. Pro-Inflammatory and Anti-Inflammatory miRNA in ACD

Concerning the involvement of miRNAs in the pathogenesis of ACD (Table 2), Vennegaard et al. were the first to describe aberrant miRNA expression in this disease, analyzing skin biopsies from subjects who had been sensitized with diphenylcyclopropenone (DPCP). They found that miRNA-21, miR-223, miR-142-3p and miR-142-5p were upregulated in the inflammatory lesions. In addition, they were also upregulated in skin biopsies from mouse models which were sensitized with 2,4-dinitrofluorobenzene, showing that mouse models are valuable tools for further study of the involvement of miRNAs in ACD [78]. Subsequently, in 2015, Gulati et al. [79] confirmed the altered expression of miR-21, miR-223, miR-142-3p and miR-142-5p in skin lesions obtained 3 and 14 days after a challenge with DPCP in sensitized subjects. They also identified 6 miRNAs that were significantly upregulated 120 days after the challenge, demonstrating the long-lasting allergen-mediated immune reaction that occurs in the skin [79]. Anderson et al. demonstrated consistent changes in miRNA expression for miR-21, miR-22, miR-27b, miR-31, miR-126, miR-155, miR-210 and miR-301a during the sensitization phase of an allergic response to TDI in a murine model [80]. In 2021, Werner et al. analyzed miRNA expression data of positive patch test reactions from patients exposed to allergens such as nickel sulphate, epoxy resin and methylochloroisothiazolinone, and to irritants such as sodium lauryl sulfate (SLS) and nonanoic acid. All allergens induced miRNAs expression in human skin, while of the irritants, only SLS did. Eighty-six miRNAs were significantly upregulated or downregulated; miR-142-3p, miR-142-5p, miR-146b and miR-155-5p were differentially expressed across all investigated allergens. Meanwhile, miR-497-5p was significantly expressed only in MCI; miR-22b-3p, miR-99a-5p, miR-193b-3p and miR-199a-3p were significantly expressed in epoxy resin ACD [81].

### 2.3. Pro-Inflammatory and Anti-Inflammatory miRNA in CSU

A few studies have examined the role of several miRNAs in the pathophysiology of CSU (Table 3). In 2017, Lin et al. studied whether miRNAs are involved in CSU regulation as biomarkers. Their study involved 12 patients who were divided according to according to whether they had active hives or no hives and to the presence or absence of CSU. MiRNAs were isolated from patient plasma, and 16 miRNAs were found to be differentially expressed in patients with active hives. Among these, miR-2355-3p, miR-4264, miR-2355-5p, miR-29c-5p and miR-361-3p were upregulated in exacerbated CSU patients; thus, they could be useful biomarkers for patients with chronic autoimmune urticaria. Then, the researchers compared these targets against urticaria-related genes. Twenty-five genes were found to match, eight of which were significantly downregulated, while the other eight were significantly upregulated.

In addition, 12 genes (Figure 2) did not serve a signaling role, whereas the other 13 were involved in regulatory pathways such as the transforming growth factor beta (TGF-β) signaling pathway (nuclear receptor subfamily 3, group C, member 1, glucocorticoid receptor (NR3C31), kit ligand (KITLG), thrombospondin I (THBS1), chemokine (C-C motif) ligand 2 (CCL2)), glucocorticoid receptor signaling pathway (NR3C1, selectin E (SELE), CCL2), p53 signaling pathway (cyclin G2 (CCNG2), THBS1, CCL2), p21-activated kinase pathway (PAK1 interacting protein 1 (PAK1IP1), KITLG, CCL2), phosphoinositide-3 kinase protein kinase B signaling pathway (KITLG, cholinergic receptors muscarinic (CHRM), THBS1) and neuroactive ligand–receptor interaction (NRC31, histamine receptors H1 (HRH1), CHRM), which could play important roles in CSU [82]. In 2019, Zhang et al. found miR-125a-5p and CCL17 (C-C motif chemokine ligand 17) to be significantly upregulated in the serum of patients with active CSU and decreased in the serum of patients in remission. Furthermore, upregulated expression of miR-125a-5p was observed in refractory CSU, indicating its potential use as a biomarker [33].

## 3. Therapeutic Perspectives

As shown in previous reports, miRNAs are involved in the pathogenesis of multiple skin disorders, including AD to CSU and ACD, making them not only potential biomarkers in disease monitoring, but also possible therapeutic targets (Figure 3).

An attempt to use miRNAs as potential biomarkers in AD was made by dosing serum miR-203 and observing that it was more elevated in the serum of patients with AD than in the controls [66]. While miR-203 may be used in AD diagnosis, miR-223 serum dosing may be helpful in AD staging, as plasma miR-223 expression levels in patients with severe AD are higher than those in patients with mild disease and healthy patients [67]. Another possible biomarker is represented by miR-483-5p, which was found to be upregulated not only in the serum of children with AD, but also in other atopic conditions, such as rhinitis and asthma, in comparison to the controls [69]. This finding could lead to the use of miR-483-5p as a biomarker of atopic predisposition.

An early therapeutic attempt was made by generating a recombinant strain of Salmonella typhimurium expressing CCL22-miRNA that was able to suppress a Th2 proinflammatory chemokine (CCL22) which is often involved in AD pathogenesis. This attempt led to the suppression of this Th2 proinflammatory cytokine and gave rise to new therapeutic approaches for this skin pathology [82]. Another potential object for therapy is miR-151a, which plays a key role in promoting the Th-2 shift [63] and, interestingly, enhancing miRNA-143, which opposes the negative regulation of filaggrin, loricrin and involucrin induced by IL-13 [73], thus potentially decreasing type-2 inflammation. Other miRNAs involved in AD inflammation that might be targets of future immunomodulatory treatments are miR-155, which decreases CTLA-4 levels and increases the proliferation of T helper cells [50,54,56], and miR-146a, a crucial regulator of factors of innate and adaptive immunity which controls immune cell differentiation, antibody production and inflammatory factor secretion [59,60]. As was thoroughly illustrated in the introduction, skin barrier impairment represents a cornerstone of AD pathophysiology. Increased miR-10a-5p expression in both the lesional and non-lesional skin of AD patients could impair keratinocyte proliferation and epidermal barrier function, directly targeting HS3, which controls keratinocyte proliferation and migration [71]. MiR-29b is also upregulated in AD lesions and is involved in keratinocyte apoptosis. Thus, these two miRNAs might be therapeutic targets for reducing skin barrier impairment [72].

However, miRNAs may have limited use as biomarkers of ACD because of their high cost and lower specificity and sensibility profiles compared to patch testing. Nevertheless, silencing specific miRNAs (miR-142-3p, miR-142-5p, miR-146b-5p miR-155-5p, miR-497-5p, miR-23b-3p, miR-99a-5p, miR-193b-3p and miR-199a-3p) [78,79,80] may be useful in the treatment of ACD due to common haptens, such as nickel sulphate, epoxy resin and methyl-chloro-isothiazolinone, when avoiding the ACD culprit is difficult for personal or professional reasons.

A few studies have evaluated miRNAs in relation to the pathophysiology and regulation of chronic urticaria. Lin et al. [82] identified 16 miRNAs (Table 1) which were particularly over-expressed during the exacerbation of CSU in 12 patients with active or non-active hives. The expression levels of five of the miRNAs (miR-2355-3p, miR-4264, miR-2355-5p, miR-29c-5p and miR361-3p) were significantly increased in the plasma samples of three patients with active hives and the presence of the Fcɛ antibody, indicating that they could be a marker for chronic autoimmune urticaria (CAU). Target prediction of these miRNAs showed their enrichment in regulatory pathways (e.g., TGF-β, glucocorticoid receptor and p53 signaling). Some other enriched terms were p21-activated kinase, phosphoinositide-3 kinase, protein kinase B and neuroactive ligand–receptor interaction [83]. Zhang et al. [33] reported upregulation of miR-125a-5p and CCL17 in CSU. Although serum levels of miR-125a-5p were even higher in refractory CSU patients, the levels were downregulated in patients who experienced remission. These results suggest that miR-125a-5p and CCL17 can serve as potential serum biomarkers of CSU, particularly the therapeutic response or disease remission. Moreover, miR-125a-5p is associated with a number of autoimmune diseases (e.g., rheumatoid arthritis, immune thrombocytopenic purpura, type 1 diabetes) [33]. A study was performed at the University of Washington (ClinicalTrials.gov identifier—NCT number: NCT02814630) to analyze and investigate the response of CSU to therapy using an miRNA array. Of particular interest was the effect of omalizumab over a three-month treatment period in adult (≥18 years) patients with CSU who remained symptomatic despite the use of high-dose H1 antihistamines. Basophil mRNA/miRNA arrays were performed to examine the role/mechanism of basophils in the immunopathogenesis of chronic urticaria. The primary outcome of the study was that miRNAs in the blood were differentially expressed after 12 weeks of treatment with omalizumab in patients with CSU, showing that specific miRNA(s) are novel biomarker(s) predicting the response to omalizumab in 20 patients with CSU. However, the data are not available. Data generated by microarray technologies, such as mRNA datasets, could be useful for identifying critical factors related to the etiopathogenesis of diseases, which could support further biological studies and potential novel therapies. Peng et al. explored immune cell infiltration and the ceRNA network, further revealing the molecular mechanism of pyroptosis-related genes in CSU and seeking to identify the potential key biomarkers. Seventeen different expressions of pyroptosis-related genes (DEPRGs) (containing four downregulated genes and thirteen upregulated genes) that are involved in inflammatory response and immunomodulation were recognized, and five hub genes (IL1B, TNF and IRF1 were upregulated; HMGB1 and P2RX7 were downregulated) were identified via the protein–protein network. DEPRGs were usually engaged in enhancing the inflammatory response, promoting interleukin-1 beta and interleukin-8 release, biological regulation (including signaling receptor activator activity, receptor ligand activity and cytokine activity) and the development of CSU. It was found that CSU tissue exhibited a higher proportion of activated mast cells, but relatively lower proportions of T cells, naïve CD4 cells, plasma cells and memory B cells.

IL1B and TNF were statistically and positively associated with activated mast cells, suggesting the maladjustment of inflammatory cells in CSU and possible immunomodulation effects. Consequently, IL1B and its related molecules might play a key role in the development of CSU, and, thus, may represent potential biomarkers of CSU. In addition, the drug–gene interaction network contained 15 potential therapeutic drugs targeting IL1B, opening up the possibility of a novel therapeutic strategy [84].

With the goal of blocking miRNAs’ activity, antagomirs and antisense oligonucleotides have been shown to be able to silence specific miRNAs in mouse models in various fields of research [85,86,87].

## 4. Conclusions

MiRNAs are short, endogenous, single-stranded non-coding RNAs that regulate genes expression post-transcriptionally [34]. As they regulate inflammatory mechanisms of the innate and adaptive immune responses, miRNAs play a key role in the pathogenesis of inflammatory skin diseases such as AD, CSU and ACD. Regarding AD, multiple studies have demonstrated that miRNAs are involved in many pathogenetic mechanisms, including immune cell maturation; regulation of the innate and adaptive immune systems [54,55,56]; suppression of Th1 cytokine expression through NFκB pathway regulation [59,60,61]; promotion of AD-related cytokines, such as IL-4, IL-17 and IL-1β [65]; and epidermal barrier function impairment targeting hyaluronan synthase 3 [71], but also the improvement of epidermal barrier function, opposing the negative regulation of filaggrin, loricrin and involucrin induced by IL-13 [75]. Regarding ACD, several studies have shown aberrant miRNA expression in skin biopsies of inflammatory lesions and altered miRNA expression in sensitized patients [78,79]. It is interesting to point out that three miRNAs (miR-146, which has an anti-inflammatory NFκB-dependent function [58,81]; miR-155, which improves skin inflammation, decreasing CTLA-4 expression [54,80]; miR-223, which is expressed by neutrophils, monocytes and eosinophils and has a proinflammatory function [66,78,79]) are found in AD and ACD pathogenesis, suggesting that some of the underlying inflammatory mechanisms might be the same for both diseases. Regarding CSU, recent studies have found that miRNAs are biomarkers of CSU activity and affect CIU pathogenesis, impacting many regulatory pathways (e.g., TGF-β, glucocorticoid receptor, p53 p21-activated kinase, phosphoinositide-3 kinase protein kinase B) [82]. It is important to point out that the altered miRNA expression in these conditions may be partially influenced by the subset of miRNAs which are constitutionally expressed in the skin. For instance, miR-21, miR-27b, miR-125a, miR-126, mir-143, miR-191, miR-199 and miR-203 are both some of the most expressed miRNAs in physiological skin and some of the main players in the pathogenesis of these diseases. Therefore, miRNAs are potential biomarkers supporting both disease diagnosis and activity monitoring. Moreover, the involvement of miRNAs in the pathogenesis of inflammatory skin diseases may make them ideal therapeutic targets. Indeed, antisense oligonucleotides, called “antagomirs”, have been found to have the ability to silence specific miRNAs in mouse models [85,86,87]. Further studies are needed in order to explore and clarify the role of miRNAs in chronic immuno-mediated dermatitis, with the aim of informing new strategies for the diagnosis, follow-up and treatment of these diseases. Key points of miRNAs’ cross-involvement in skin allergies are described in Table 4.

## Figures and Tables

**Figure 1 biomedicines-11-01266-f001:**
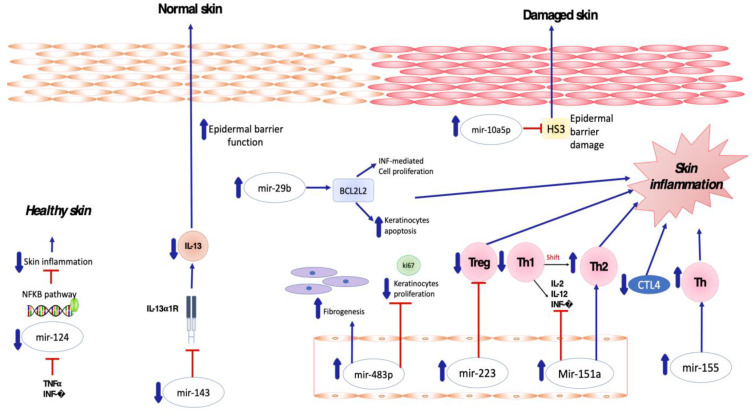
Skewed skin inflammation associated with dysregulated miRNA expression in patients with Atopic Dermatitis.

**Figure 2 biomedicines-11-01266-f002:**
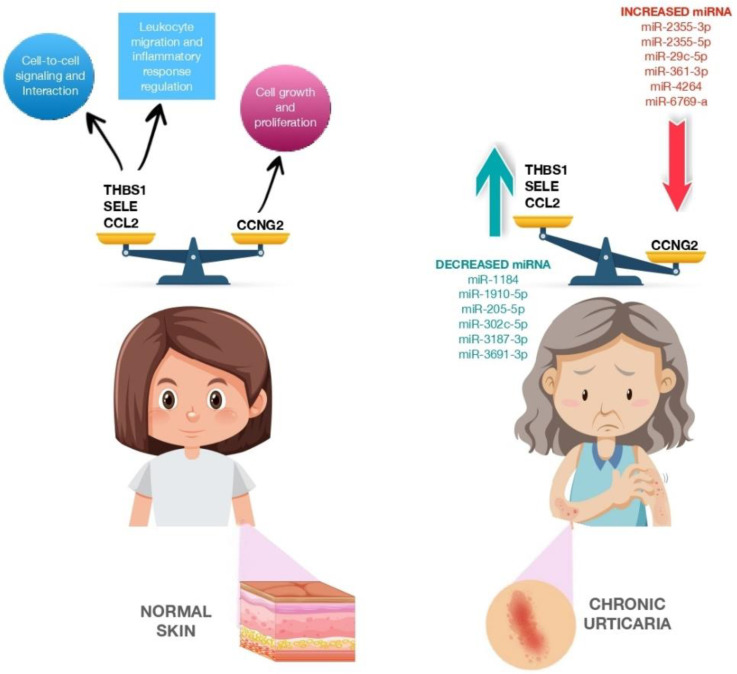
CCNG2, THBS1, SELE and CCL2 expression in normal skin and CCNG2, THBS1, SELE and CCL2 expression modulated by miRNA dysregulation in patients with chronic urticaria. The figure is designed by Freepik.

**Figure 3 biomedicines-11-01266-f003:**
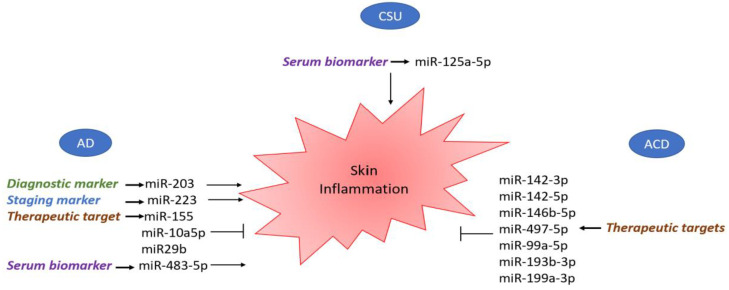
Skin inflammation modulated by miRNA expression in atopic dermatitis, chronic spontaneous urticaria and allergic contact dermatitis. The figure is designed by Freepik.

**Table 1 biomedicines-11-01266-t001:** miRNA in atopic dermatitis. (CTLA-4: cytotoxic T-lymphocyte antigen 4; CCL5: chemokine (C-C motif) ligand 5; CCL8: C-C motif chemokine ligand 8; IRAK-1: interleukin 1 receptor-associated kinase 1; CARD-10: caspase recruitment domain-containing protein 10; HAS3: hyaluronan synthase 3.

Study	miRNA	Study Population	Samples	Results	Molecular Pathways/Mechanism of Action
Sonkoly, E., et al. [54] 2010	miR-155	Humans (18)	Serum	Upregulated	Overexpression of miR-155 decreases CTLA-4 levels and increases proliferation in T helper cells, promoting chronic skin inflammation.
Rebane, A., et al. [58] 2014	miR-146a	Humans/mice	Skin	Upregulated	miR-146a decreases the expression of IFN-γ-inducible genes CCL5, CCL8 and ubiquitin D (UBD) in keratinocytes and in a mouse model of AD by targeting the upstream mediators of NF-κB signaling—IRAK1 and CARD10.
Chen, X.F., et al. [63] 2014	miR-151a	Humans (500)	Plasma	Overexpressed	miR-151a targets the IL-12 receptor β2 (IL12RB2), a subunit of the IL-12 receptor.
Jia, H.Z., et al. [66] 2018	miR-223	N/A	Serum	Upregulated	miR-223 expression is correlated with lower Treg cell numbers, a decreased number of which at birth correlates with an increased risk of AD.
Lv, Y., et al. [49] 2014	miR-483-5p	Humans (30)	Serum and urine	Upregulated	miR-483-5p modulates fibrogenesis through the regulation of collagen homeostasis.
Vaher, H., et al. [51] 2019	miR-10a-5p	Humans (10)	Skin (lesional and non-lesional)	Upregulated	miR-10a-5p is a direct target of HAS3, a damage-associated positive regulator of keratinocyte proliferation and migration. Upregulation of miR-10a-5p affects keratinocyte proliferation, thus impairing normal skin barrier function.
Gu, C., et al. [52] 2017	miR-29b	Humans (21)	Lesional skin and serum	Upregulated	miR-29b triggers IFN-γ-mediated apoptosis of keratinocytes by targeting BCL2L.
Lv, Y., et al. [69] 2014	miR-203	Humans (30)	Serum	Upregulated	The miR-203 target gene is the regulator of cytokine production SOCS-3 (suppressor of cytokine signaling 3).
			Skin	Upregulated	
			Urine	Downregulated	
Zeng, Y.P., et al. [75] 2018	miR-143	N/A	Skin	Downregulated	miRNA-143 decreases IL-13 activity and inflammatory reaction by inhibiting IL-13 receptor-alpha1 (IL-13Ra1) in epidermal keratinocytes.
Yang, Z., et al. [77] 2017	miR-124	Humans (37)	Serum	Downregulated	miR-124 inhibits the p65 subunit of NF-kB and downregulates CCL5 and CCL8, thereby regulating inflammatory responses of keratinocytes and chronic skin inflammation in AD.

**Table 2 biomedicines-11-01266-t002:** miRNA in allergic contact dermatitis. (TGF-β: transforming growth factor beta; SMAD3: mothers against decapentaplegic homolog 3; TGIF1: TGFB-induced factor homeobox 1; IGF3: insulin-like growth factor-3).

Study	miRNA	Study Population	Samples	Results	Molecular Pathways/Mechanism of Action
Werner et al. [81] 2020	miR-142-3p, miR-142-5p, miR-146b-5p, miR-155-5p	Humans (nickel sulfate, epoxy resin (EP) and methylochloroisothia zolinone (MCI); *n* = 5 for each), irritants (sodium lauryl sulfate (SLS, *n* = 9) and nonanoic acid (NO, *n* = 5)) and from non-affected skin (baseline, *n* = 5).	Skin	Upregulated	miR-155-5p: enrichment of biological processes for axon guidance, smooth muscle cell migration and leukocyte/T cell apoptotic process.
Werner et al. [81] 2020	miR-497-5p	Humans (nickel sulfate, epoxy resin (EP) and methylochloroisothia zolinone (MCI); *n* = 5 for each), irritants (sodium lauryl sulfate (SLS, *n* = 9) and nonanoic acid (NO, *n* = 5)) and from non-affected skin (baseline, *n* = 5).	Skin (patch tests with MCI)	Upregulated	T cell activation, cell–cell adhesion, cytokine and chemokine regulation pathways and a role in TGF-β-pathways via the regulation of SMAD3.
Werner et al. [81] 2020	miR-23b-3p, miR-99a-5p, miR-193b-3p, miR-199a-3p	Humans (nickel sulfate, epoxy resin (EP) and methylochloroisothia zolinone (MCI); *n* = 5 for each), irritants (sodium lauryl sulfate (SLS, *n* = 9) and nonanoic acid (NO, *n* = 5)) and from non-affected skin (baseline, *n* = 5).	Skin (Patch Tests with MCI)	Upregulated	miR23b-3p and miR-99a-5p: skin homeostasis and development in vitro via TGIF1 and IGFR1. miR-193b-3p and miR-199a-3: leukocyte proliferation and keratinocyte/epidermis differentiation.
Vennegaard et al. [78] 2012	miR-21, miR-223, miR-142-3p, miR-142-5p	Humans (nickel sulfate, epoxy resin (EP) and methylochloroisothia zolinone (MCI); *n* = 5 for each), irritants (sodium lauryl sulfate (SLS, *n* = 9) and nonanoic acid (NO, *n* = 5)) and from non-affected skin (baseline, *n* = 5).	Skin	Upregulated	T cells, T cell activation and skin inflammation.
Gulati et al. [79] 2015	miR-21, miR-223, miR-142-3p, miR-142-5p	Humans (7) (DPCP at day 3, day 14 and day 120)	Skin	Upregulated	T cells, T cell activation and skin inflammation.
Anderson et al. [80] 2014	miR-21, miR-22, miR-155, miR-126, miR-27b, miR-210, miR-31, miR-301a	Murine (toluene 2,4-diisocyanate (TDI)).	Skin	Upregulated	T cells, T cell activation and skin inflammation.

**Table 3 biomedicines-11-01266-t003:** miRNA in chronic spontaneous urticaria. (BLC2: B-cell lymphoma 2; STAT3: signal transducer and activator of transcription 3; TGF-β: transforming growth factor beta; CCL17: C-C motif chemokine ligand 17).

Study	miRNA	Study Population	Samples	Results	Molecular Pathways/Mechanism of Action
Lin et al. [82] 2017	miR-2355-3p miR-2355-5p miR-4264 miR-29c-5p miR-361-3p miR-6769a-5p	Humans (12)	Serum	Upregulated	Cell growth and proliferation
Lin et al. [82] 2017	miR-1184 miR-1910-5p miR-205-5p miR-302c-5p miR-3187-3p miR-3691-3p miR-4649-5p miR-4733-5p miR-6799-3p miR-6800-3p	Humans (12)	Serum	Downregulated	Cell-to-cell signaling and interaction, cellular movement, regulation of leukocyte migration, tissue development immune cell trafficking, regulation of inflammatory response
Zhang et al. [33] 2019	miR-125a-5p	Humans (20 active CIU patients and 20 healthy controls)	Serum	Upregulated	BLC2, STAT3, TGF-β and CCL17

**Table 4 biomedicines-11-01266-t004:** Key points of miRNAs’ cross-involvement in skin allergies.

Key Points
miRNAs are short, non-coding RNAs which regulate gene expression post-transcriptionally and have a key role in the pathogenesis of inflammatory skin diseases such as AD, CSU and ACD;
miRNAs are involved in many pathways affecting the function of the innate and adaptive immune systems and the skin barrier integrity in AD;
miRNAs are found to be aberrated or altered in inflammatory lesions of ACD;
Some miRNAs are involved in the pathogenesis of both AD and ACD;
Several miRNAs are isolated in CSU exacerbations, and others impact many pathways involved in CIU pathogenesis;
miRNAs may become biomarkers for diagnosis and follow-up in inflammatory skin diseases in the future;
miRNAs may become therapeutic targets in inflammatory skin diseases in the future.

## Data Availability

Data supporting the review can be found at pubmed.

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
