# Peer review of "miRNAs’ Cross-Involvement in Skin Allergies: A New Horizon for the Pathogenesis, Diagnosis and Therapy of Atopic Dermatitis, Allergic Contact Dermatitis and Chronic Spontaneous Urticaria"

_biomedicines, 2023, doi:10.3390/biomedicines11051266_

Round 1
Reviewer 1 Report
Dear authors,
I suggest that you make some corrections to improve the manuscript:
lines 142-143 please add some information about the mechanism of circRNA regulation of miRNA levels.
Please add some words about the fact that one miRNA can influence up to hundreds genes expression.
Please add some sentences about other types of epigenetic gene expression regulation like DNA hypermethylation and acetylation, at least related AD pathogenesis
line 324 'miR NAs' two words should be united,
line 395 'nflamatory' change to 'inflammatory'.
Author Response
Dear Reviewer, I revise the paper accordingly to your suggestions.
Giuseppe Murdaca

Reviewer 2 Report
In the present review, the authors have reviewed the involvement of miRNAs in the etiopathogenesis of three skin disorders: atopic dermatitis, allergic contact dermatitis and chronic spontaneous urticaria.
The review is well-written with only some minor stylistic flaws.
Suggestion for improvements:
-Are there any miRNAs commonly involved in two or three of these disorders? Any reference addressing this question?
-The pattern of altered miRNA expression could be also partly dependent on the tissue-specific subset of miRNA that are physiologically expressed in the skin. The authors could elaborate on this point.
Minor
-Lines 151-160 should be separated from paragraph 1.4 and put under a different subheading
-Paragraph 2.1 should be divided into three parts, one for each skin disorder
Author Response
Dear reviewer, I revisae the paper accordingly to your suggestions.
Giuseppe Murdaca
